

# Demographics and density estimates of two three-toed box turtle (*Terrapene carolina triunguis*) populations within forest and restored prairie sites in central Missouri

Kelly M. O'Connor[1], Chadwick D. Rittenhouse[1], Joshua J. Millspaugh[2] and Tracy A.G. Rittenhouse[1]

[1] Wildlife and Fisheries Conservation Center, Department of Natural Resources and the Environment, University of Connecticut, Storrs, CT, USA
[2] Department of Fisheries and Wildlife Sciences, University of Missouri, Columbia, MO, USA

## ABSTRACT

Box turtles (*Terrapene carolina*) are widely distributed but vulnerable to population decline across their range. Using distance sampling, morphometric data, and an index of carapace damage, we surveyed three-toed box turtles (*Terrapene carolina triunguis*) at 2 sites in central Missouri, and compared differences in detection probabilities when transects were walked by one or two observers. Our estimated turtle densities within forested cover was less at the Thomas S. Baskett Wildlife Research and Education Center, a site dominated by eastern hardwood forest ($d = 1.85$ turtles/ha, 95% CI [1.13, 3.03]) than at the Prairie Fork Conservation Area, a site containing a mix of open field and hardwood forest ($d = 4.14$ turtles/ha, 95% CI [1.99, 8.62]). Turtles at Baskett were significantly older and larger than turtles at Prairie Fork. Damage to the carapace did not differ significantly between the 2 populations despite the more prevalent habitat management including mowing and prescribed fire at Prairie Fork. We achieved improved estimates of density using two rather than one observer at Prairie Fork, but negligible differences in density estimates between the two methods at Baskett. Error associated with probability of detection decreased at both sites with the addition of a second observer. We provide demographic data on three-toed box turtles that suggest the use of a range of habitat conditions by three-toed box turtles. This case study suggests that habitat management practices and their impacts on habitat composition may be a cause of the differences observed in our focal populations of turtles.

## INTRODUCTION

Anthropogenic land-use and land-cover change are major drivers of species decline globally (*Fischer & Lindenmayer, 2007*). In the central United States, intensification of agricultural practices has been linked to species declines across taxa (*Samson, Knopf &*

Corresponding author
Kelly M. O'Connor,
kelly.oconnor@uconn.edu

*Ostlie, 2004*). Three-toed box turtles (*Terrapene c. triunguis*) in this region use mixed forest and open field habitat to fulfill their life history requirements (*Reagan, 1974*; *Schwartz & Schwartz, 1991*). Habitat suitable to three-toed box turtles may coincide with areas undergoing management for prairie restoration, maintenance of old field habitats, or creation of agricultural areas.

Habitat modification due to changing land-use can negatively affect populations of box turtles through altered movement and behavior, increased rates of mortality, and reduced genetic diversity through increased isolation. Daily movements of eastern box turtles (*Terrapene carolina carolina*) decreased in distance and increased in frequency following timber harvest (*Currylow, Macgowan & Williams, 2012*). Population structure may be altered as nesting females and dispersing juveniles experience higher rates of road mortality in a heavily fragmented landscape (*Steen & Gibbs, 2004*). Nests in close proximity to edge habitat are often more heavily depredated than those located a greater distance from fragmented edges (*Temple, 1987*; *Shake, Moorman & Burchell II, 2011*). Turtle populations isolated as a result of habitat fragmentation may experience a lack of genetic diversity, reducing the long-term viability of a given population (*Kuo & Janzen, 2004*; *Richtsmeier et al., 2008*; *Marsack & Swanson, 2009*).

Management practices to maintain prairie and old field habitat, or to convert other habitat to agricultural uses, are known to be harmful to turtles in certain instances. Serious injuries may occur where turtles come in contact with agricultural machinery (*Saumure & Bider, 1998*; *Nazdrowicz, Bowman & Roth, 2008*). Twenty percent of a population of eastern box turtles occurring in an area of frequent prescribed fire showed lasting injuries caused by fire and weighed less at a given length than turtles in unburned areas (*Howey & Roosenburg, 2013*). Although management activities can result in undesired consequences including harm to individual animals, active management is needed to maintain prairie and old field habitat and thus wildlife populations that utilize these habitats. Managers are left attempting to minimize negative consequences of otherwise beneficial management activities, and small modifications to the implementation of management actions can be important.

Demographic data on populations of box turtles wherever they occur are valuable in their ability to inform relevant conservation actions, and may be particularly relevant when attempting to monitor changes in populations and to assess the need for conservation actions as a result of habitat alteration. The age structure of a population may indicate ongoing growth or decline (*Alexander, 1958*). Skewed sex ratios impact the effective size of a population, which in turn may alter recruitment and age structure (*Browne & Hecnar, 2007*). Accuracy in estimating abundance is critical when attempting to detect changes in populations of cryptic species with low probabilities of detection (*Couturier et al., 2013*). Box turtles are good candidates for surveying using distance sampling methods as they are slow moving and unlikely to immediately travel large distances in response to observer movements along a transect. However, box turtles may be difficult to detect when levels of activity are low or when turtles are located in dense vegetative cover (*Refsnider et al., 2011*). A comparison of single observer, double observer, and total count methods

used to survey gopher tortoises (*Gopherus polyphemus*) reported an exact match between burrow abundance estimates calculated using double observer methodology and a total count of burrows (*Nomani, Carthy & Oli, 2008*). Double observer line transect studies are, however, more labor intensive and, thus, more costly to conduct. Two observers may be unnecessary if detection probabilities are not significantly improved over those achieved by one observer.

Many studies monitoring box turtle populations have focused on populations of turtles occurring on protected lands lacking differences in land use and land cover (*Stickel, 1978*; *Williams & Parker, 1987*; *Garber & Burger, 1995*). Our goal is to describe two populations of three-toed box turtles at sites featuring varying degrees of mixed hardwood forest and open field habitat, as well as differences in habitat management practices. Our specific objectives are to determine the densities of turtles within forested areas of each study site, to test for differences in age structure between populations using morphometric data, and to quantify severity of injuries to turtle shells. We also report single versus double observer study designs for estimating box turtle density using distance sampling to inform future monitoring efforts.

## MATERIALS AND METHODS

### Study area

We collected data at the Thomas S. Baskett Wildlife Research and Education Center (Baskett) and the Prairie Fork Conservation Area (Prairie Fork) in central Missouri (Fig. 1). Both Baskett and Prairie Fork have been agricultural land with both pasture and crop production within the past century. Baskett (38.7735°N, 92.1975°W) totals 917 ha of land, 67% of which is second-growth forest deciduous forest. Oak (*Quercus* sp.) and hickory (*Carya* sp.) dominate the overstory and sugar maple (*Acer saccharum)* dominates the understory. Coniferous forest accounts for approximately 22% of the total site. This site has received minimal management since its purchase by the state in the late 1930s. Approximately 9% of the total site, both in the north end and outside of the area sampled for turtles, is comprised of open old field habitat. This is maintained through prescribed fire and mowing every 1–3 years. The last instance of timber harvest in Baskett was in the late 1980s and affected approximately 2% of all forested land (*Thompson & Fritzell, 1989*).

Prairie Fork (38.8929°N, 91.7331°W) totals 290 ha comprised primarily of restored tallgrass prairie and old field (68%). Patches of oak (*Quercus* sp.) and hickory (*Carya* sp.) forest occur along the small streams, and a large continuous tract of forest is located southwest of the site. Deciduous forest cover accounts for about 30% of the property, with an additional 1% classified as coniferous forest cover. Restoration efforts began in 1996 to convert agricultural land to prairie or old field. Mowing and prescribed fire occur somewhere on the site every year, with 2% of land managed as food plots. Mowing is also used 2–4 times per month to maintain an extensive network of trails that support the educational, recreational, and research-oriented activities at Prairie Fork.

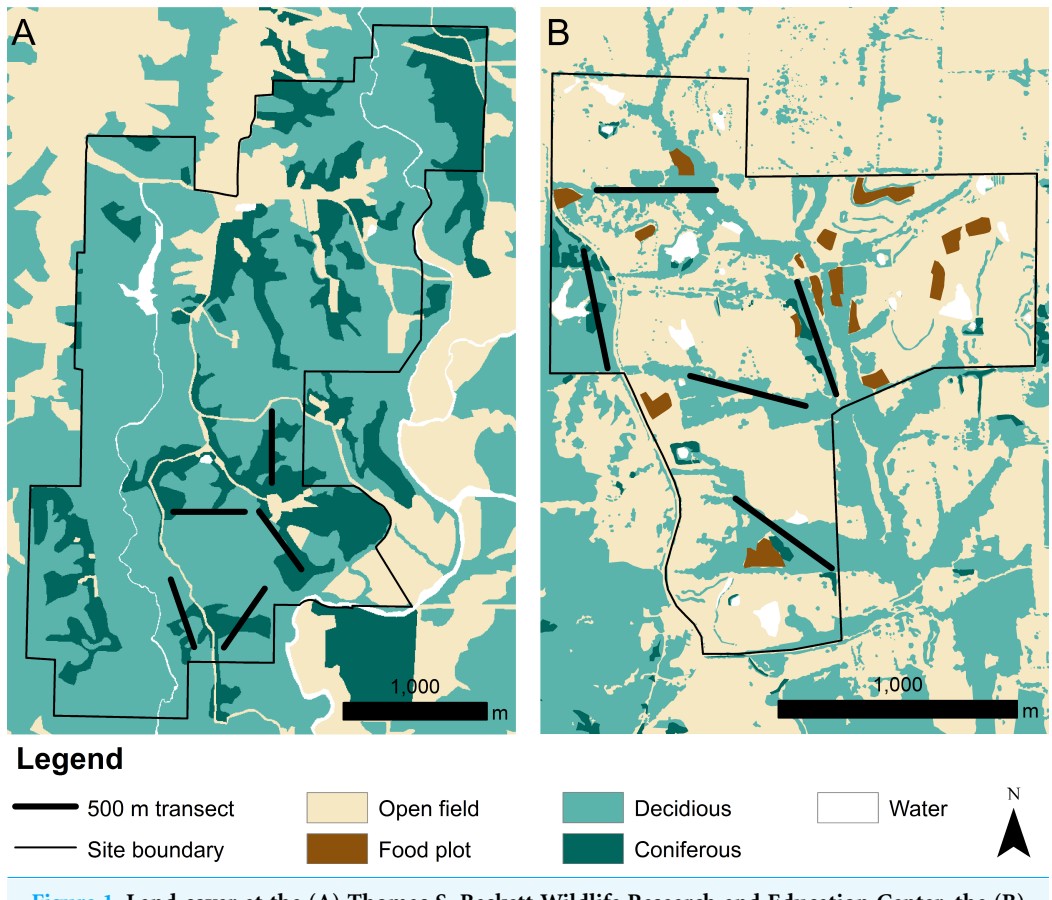

**Legend**

| | | | | | | | |
|---|---|---|---|---|---|---|---|
| ▬▬ | 500 m transect | ▢ | Open field | ▢ | Deciduous | ▢ | Water |
| ── | Site boundary | ▢ | Food plot | ▢ | Coniferous | | |

**Figure 1** **Land cover at the (A) Thomas S. Baskett Wildlife Research and Education Center, the (B) Prairie Fork Conservation Area, and surrounding areas.**

## Data collection

We used distance sampling methodology to survey our study sites for box turtles, specifically line transect sampling. Observed distances perpendicular from a transect are used to calculate probability functions modelling the decrease in detection probabilities of animals with distance from central transect line, and in turn this function is used to estimate overall density of a species within a surveyed area (*Buckland et al., 2001*). The likelihood of detecting individuals often varies among habitat types, with detection probability decreasing more rapidly with distance from the transect in dense habitats compared to open habitats. See *Buckland et al. (2001)* for a detailed explanation of line transect sampling. We conducted surveys for box turtles along five 500 m line transects spaced a minimum of 100 m apart at each study site. We used guidelines presented by *Buckland et al. (2001)* when selecting the length of transect, where transect length is based on a desired level of precision in the estimate of density and a predicted encounter rate. We established transects in forested habitats in both study areas. Transect placement was based on the distance sampling assumption that objects directly on the transect must be detected. This assumption would almost certainly be violated in densely vegetated old field and prairie habitat. Therefore, our density estimates apply only within the forested areas of

our study sites. Transects at Baskett were placed within a 290 ha sample area comparable to the total size of Prairie Fork. Transects were stratified across the site to minimize travel time between transects while maintaining a 100 m buffer between two transects and allowing for adequate spatial coverage of the site. Our study sites were not large enough to use the 10–20 replicate transects suggested by *Buckland et al. (2001)*. Instead we achieved the desired total transect length by revisiting transects repeatedly over time. Transects were surveyed the same day of the week each week, and were only postponed for heavy rain. Transects were walked in the morning to avoid sampling during the late afternoon peak in daily temperature when turtle activity may be reduced. The starting transect of surveys was varied at random each week to ensure transects were sampled at different times within the morning. The survey protocol followed a two observer design with one observer remaining on the line and the second crossing over the line as a sweeper (*Buckland et al., 2001*). Sweepers attempted to standardize distance travelled from a transect to approximately 20 m. For all turtles found we recorded the perpendicular distance (m) from the turtle to the line transect as well as the position of the observer ("on the line" or "sweeper"). We sampled transects weekly between 14 May and 30 July 2007 for a total of 12 visits per transect per site.

All captured turtles received a series of notches along the marginal scutes to serve as an identification number (*Cagle, 1939*; *Schwartz & Schwartz, 1974*). We recorded an estimate of minimum age of each turtle as determined by the number of distinguishable annuli on dorsal scutes. We also assigned turtles to age classes 1, 2, or 3 based on the number and condition of annuli, overall coloration, and carapace length such that age class 1 = juvenile (<110 mm carapace length with discernible annuli), age class 2 = adult (>110 mm carapace length with discernible annuli), and age class 3 = old adult (>110 mm carapace length with heavily worn or no discernible annuli) (*Schwartz & Schwartz, 1974*). We used calipers to measure: (1) maximum carapace length; (2) maximum carapace width; (3) plastral hinge width; (4) maximum carapace height; (5) plastron anterior length; (6) plastron posterior length; and (7) plastron total length. We recorded the mass of each turtle using a spring scale. We also measured and recorded damage to the carapace in the form of a carapace mutilation index (CMI) (*Saumure, Herman & Titman, 2007*). We scored damage on a quadrat basis ranging from a score of zero (intact) to three (severe) and divided by a maximum possible score of 12 in order to create an index value (*Saumure, Herman & Titman, 2007*). The same individual was responsible for all CMI scoring throughout the duration of the study to ensure consistency in score assignments. University of Missouri Animal Care and Use Committee approved this research (Protocols 3629 and 4291).

## Density estimation

Distance 6.1 (*Thomas et al., 2010*) was used to estimate turtle density in forested areas of Prairie Fork and Baskett. We assumed that population density was constant during our study period, and pooled data from repeated visits to transects for the analysis. Outlying observation distances were determined using a visual histogram analysis of the data and outlying values were truncated to address monotonicity, as is recommended in *Buckland et al. (2001)*. We fit 4 detection function models to each data set: (1) half normal cosine;

(2) hazard rate polynomial; (3) uniform cosine; and (4) half normal hermite polynomial and ranked model performance according to Akaike Information Criterion corrected for small sample sizes (AICc) (*Burnham & Anderson, 2002*). These models are all suggested as generally useful models of detection functions that may be broadly applicable to distance sampling datasets where appropriate truncation has occurred (*Buckland et al., 2001*). We repeated the process of model building and comparison including observations for the "line" observer only (*Thomas et al., 2010*). We truncated data at the same distances in the single observer models as in the double observer models and restricted both the Baskett and Prairie Fork models to two orders of adjustment to address issues of monotonicity.

### Analysis of morphometric measurements

We used a two sample t-test to determine if morphological measurements differed significantly between the Baskett and Prairie Fork data sets. We tested nonparametric data sets using a Mann–Whitney U test. All tests were conducted with $\alpha = 0.05$. We used SAS 9.3 (SAS Institute Inc., Cary, North Carolina, USA) for all statistical analyses. We used PROC GLM to perform an ANCOVA testing the effects of minimum age, site, and the interaction of the two on variation in carapace maximum length. We assessed morphometric data for normality through visual analysis of histogram and boxplot charts and followed up on visual analyses with statistical tests of normality using PROC Univariate in SAS. Turtles that were truncated from distance analyses were still included as data points in analyses of morphometric measurements. Carapace height, plastron anterior length, plastron posterior length, plastron total length, and carapace mutilation index values did not qualify as normally distributed. The remaining morphometric measurements were normally distributed.

## RESULTS

### Density estimation

We captured 51 individual turtles along five transects over the two and a half month sampling period at Baskett. We captured 55 individual turtles along five transects at Prairie Fork. Seven turtles were recaptured at least once at both Baskett and Prairie Fork. The maximum number of recaptures for an individual turtle was four at Baskett and three at Prairie Fork. Our approximate encounter rate across both of our study sites was 10.8 turtles/km. For the Baskett data the top ranked model with no models within two AIC units was the hazard rate polynomial (AIC$_c$ = 649.97, $\omega$ = 0.69). The hazard rate polynomial model produced an average probability of detection within the Baskett search area of $P = 0.77$ (Fig. 2A) and an estimated density of 1.85 turtles/ha (Table 1). For Prairie Fork there were three models within two AIC units of one another. We chose to use the hazard rate polynomial function (AIC$_c$ = 545.11, $\omega$ = 0.33) as it was the best supported model fit to the Baskett data set and was one of a set of models within two AIC units of each other for the Prairie Fork data set. The estimated average probability of detection within the Prairie Fork search area was $P = 0.43$ (Fig. 2C) and the density of turtles was 4.14 turtles/ha (Table 1).

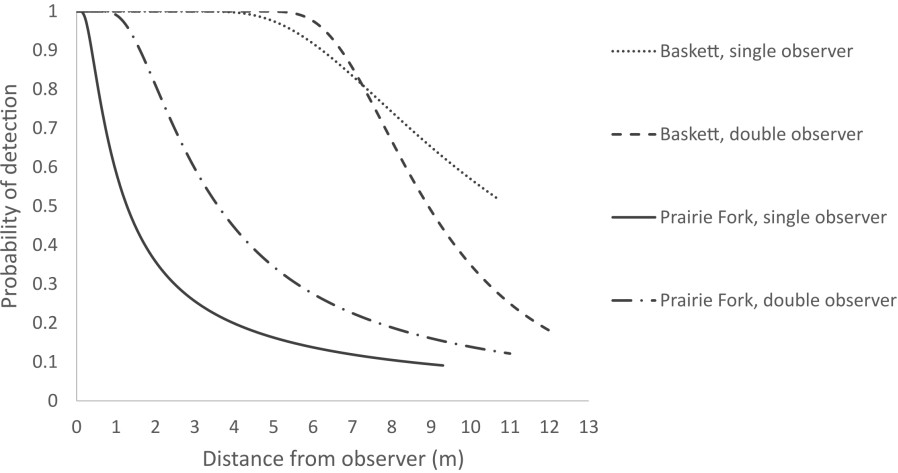

**Figure 2** Single and double observer detection probabilities for the Thomas S. Baskett Wildlife Research and Education Center (Baskett) and the Prairie Fork Conservation Area (Prairie Fork).

**Table 1** Probability of detection (*P*) and density estimates in number of turtles per hectare (*D*) for three-toed box turtle populations sampled at the Thomas S. Baskett Wildlife Research and Education Center (BA) and the Prairie Fork Conservation Area (PF).

| Study area | Observers | Observations | Average distance to turtle (m) | SD | *P* | SE | 95% CI | *D* (turtles/ha) | SE | 95% CI |
|---|---|---|---|---|---|---|---|---|---|---|
| BA | 1 | 38 | 4.90 | 2.97 | 0.87 | 0.15 | (0.62, 0.99) | 0.68 | 0.15 | (0.43, 1.09) |
| | 2 | 102 | 6.23 | 4.25 | 0.77 | 0.06 | (0.66, 0.89) | 1.85 | 0.37 | (1.13, 3.03) |
| PF | 1 | 37 | 2.83 | 2.72 | 0.27 | 0.10 | (0.13, 0.56) | 3.36 | 1.70 | (1.19, 9.49) |
| | 2 | 108 | 3.53 | 2.97 | 0.43 | 0.07 | (0.32, 0.59) | 4.14 | 1.32 | (1.99, 8.62) |

We selected a hazard rate polynomial model (AIC$_c$ = 327.11, $\omega$ = 0.33) to fit a single observer analysis of the Baskett data (Fig. 2B). We chose this model to maintain consistency in model selection among our other data analyses as it was again one of multiple competing models. Estimated density of turtles within the Baskett search area was approximately 0.68 turtles/ha with an average probability of detection of *P* = 0.87 (Table 1). We fitted the single observer Prairie Fork data to a hazard rate polynomial function (AIC$_c$ = 247.7, $\omega$ = 0.27) to maintain consistency in model selection where multiple models were competing (Fig. 2D). Estimated density of turtles within the Prairie Fork search area was 3.36 turtles/ha with an average probability of detection of *P* = 0.27 (Table 1).

## Analysis of morphometric measurements

All morphometric measurements with the exception of carapace mutilation were significantly larger for turtles sampled at Baskett than at Prairie Fork (Table 2). Carapace mutilation index values did not differ significantly between turtles captured at Baskett and Prairie Fork (Table 2 and Fig. 3). Turtles at Baskett were also significantly older than turtles at Prairie Fork (Table 2 and Fig. 4). Juveniles accounted for approximately 9.5% of the population at Prairie Fork and 3.1% of the population at Baskett. We observed

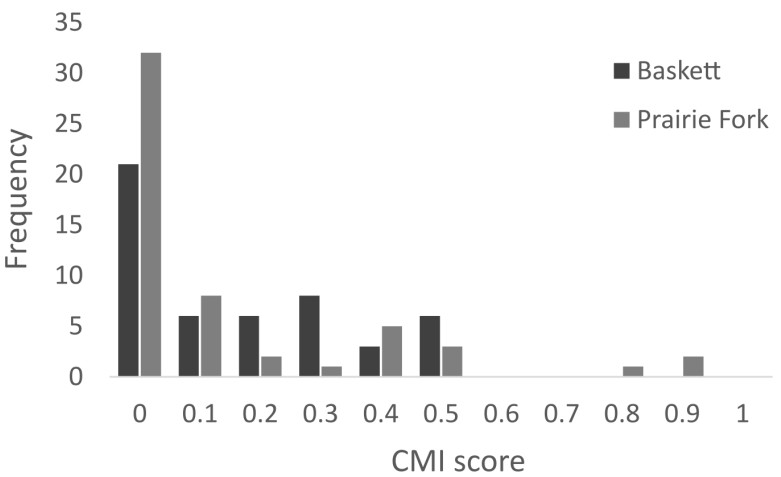

**Figure 3 Carapace mutilation index (CMI) scores of turtles sampled at the Thomas S. Baskett Wildlife Research and Education Center (Baskett) and the Prairie Fork Conservation Area (Prairie Fork).**

**Table 2 Significance tests for differences between morphometric measurements in captured turtles at the Thomas S. Baskett Wildlife Research and Education Center (BA) and the Prairie Fork Conservation Area (PF).**

|  | $n$ (BA) | BA (mean, SD) | $n$ (PF) | PF (mean, SD) | $P$ value |
|---|---|---|---|---|---|
| Minimum age (Years) | 43 | (11.36, 2.97) | 51 | (9.81, 2.97) | 0.008 |
| Age class | 50 | (2.1, 0.61) | 54 | (1.9, 0.66) | 0.05 |
| Carapace max. length (mm) | 50 | (137.4, 14.9) | 54 | (129.1, 17.5) | 0.005 |
| Carapace max. width (mm) | 50 | (106, 11.5) | 54 | (99, 12.7) | 0.002 |
| Plastral hinge width (mm) | 50 | (100.2, 10.9) | 54 | (93.9, 11.6) | 0.002 |
| Carapace max height (mm) | 50 | (68.57, 7.47) | 54 | (64.38, 8.37) | 0.004 |
| Plastron anterior length (mm) | 50 | (56.1, 6.08) | 54 | (52.06, 6.76) | 0.0003 |
| Plastron posterior length (mm) | 50 | (80.65, 8.52) | 54 | (75.3, 10.5) | 0.005 |
| Plastron total length (mm) | 50 | (136.4, 14.6) | 54 | (127.9, 16.8) | 0.0019 |
| Mass (g) | 50 | (544, 151) | 54 | (460, 154) | 0.003 |
| CMI | 50 | (0.142, 0.154) | 54 | (0.123, 0.215) | 0.126 |

a male to female ratio of 1:1 at Baskett and 2.38:1 at Prairie Fork. The breakdown of age classes of captured turtles for both sites is presented in Fig. 5. For two turtles annuli could not be accurately counted due to excessive wear to the carapace. We assigned these individuals to age class 3. In our ANCOVA, the minimum age covariate was significantly related to maximum carapace length ($F = 168.8, p \leq 0.00001$), but neither our site factor ($F = 0.37, p = 0.54$), nor the interaction of site and minimum age ($F = 2.95, p = 0.09$) were significantly related to maximum carapace length.

## DISCUSSION

Box turtles (*Terrapene carolina*) are a widely distributed species in the Eastern United States but are considered vulnerable to an array of threats throughout their range. Data

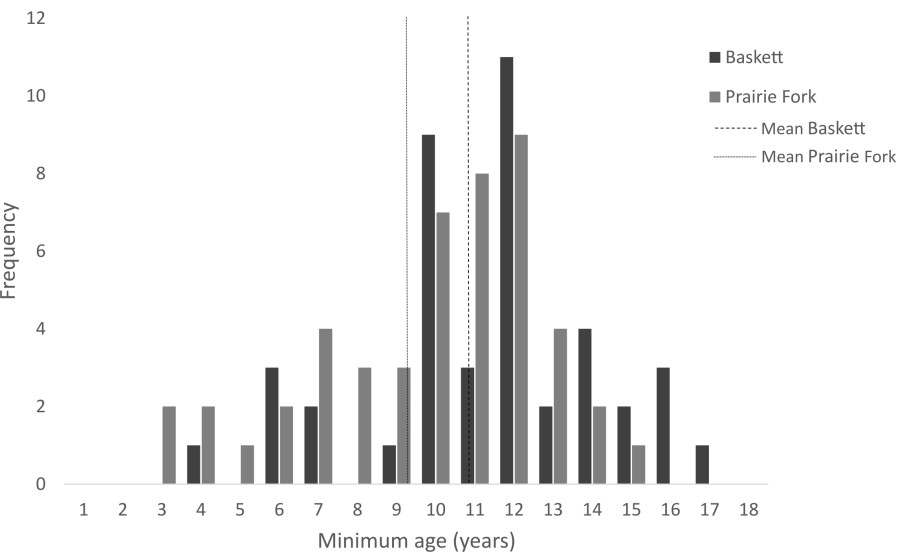

**Figure 4** Minimum age of turtles at the Thomas S. Baskett Wildlife Research and Education Center (Baskett) and the Prairie Fork Conservation Area (Prairie Fork).

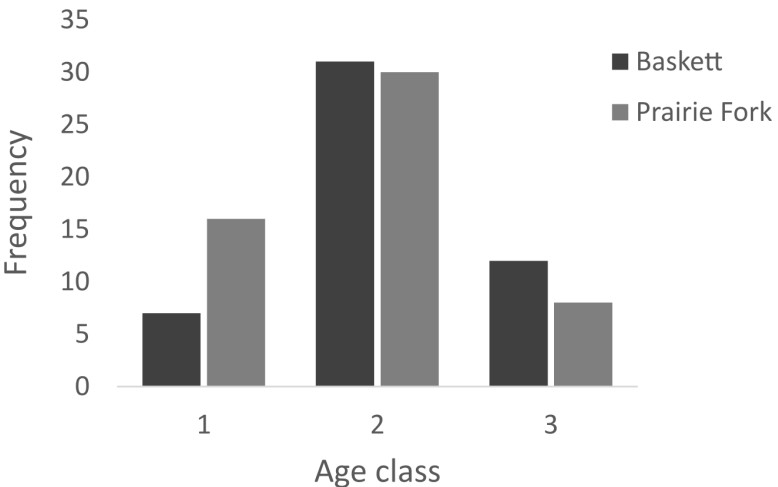

**Figure 5** Number of individuals in age classes 1, 2, and 3 at the Thomas S. Baskett Wildlife Research and Education Center (Baskett) and the Prairie Fork Conservation Area (Prairie Fork).

on distinct populations are needed to assess the need for and to direct appropriate conservations actions in the future. We present demographic information of two populations of three-toed box turtles in central Missouri. Density of three-toed box turtles in forested areas was lower at Baskett than at Prairie Fork. This case study suggests that management practices and their impacts on habitat composition may be a potential source of the differences we observed in our studied box turtle populations.

Where Baskett is dominated by uniform forest, Prairie Fork is a mosaic of prairie and forest habitat. We observed a greater proportion of juveniles at Prairie Fork than at Baskett which may be a result of differences in continuity of canopy cover at these two

sites. Eastern box turtle nesting habitat has been shown to have less woody vegetation, less leaf litter, more bare ground, less canopy cover, and higher light intensity when compared to random sites (*Flitz & Mullin, 2006*). Conversely, the opposite has been observed in Florida box turtle (*Terrapene carolina bauri*), where female turtle habitat use did not change significantly based on season (*Dodd, Franz & Smith, 1994*). This study focused only on used habitats and substrates, and did not make measured comparisons to randomly available habitat. All but one of the female eastern box turtles radio monitored by *Nazdrowicz, Bowman & Roth (2008)* nested in large, open fields. We suggest the large tract of continuous, mature second-growth oak–hickory forest with dense sugar maple understory found at Baskett may be limited in habitat suitable for nesting. Prairie Fork contains a greater proportion of open field habitat that may provide resident turtles with greater nesting opportunities resulting in relatively more young turtles at this site.

We estimated the density of three-toed box turtles in forested areas was greater at Prairie Fork (4.14 turtles/ha) than at Baskett (1.85 turtles/ha). These estimates of density are similar to those of *Williams & Parker (1987)*, who estimated 4.4–5.7 box turtles/ha in their study site in the Allee Memorial Woods in Indiana. Our estimates of turtle density are greater than density estimates of eastern box turtle populations in a fragmented landscape in Delaware where human activity is prominent (adult density = 0.81–0.86 turtles/ha) (*Nazdrowicz, Bowman & Roth, 2008*). Those estimates were collected in the Eastern United States where turtle demographics may be more heavily influenced by interaction with humans and high rates of urban and suburban development. A large forested area in North Carolina contained densities of eastern box turtles (density = 1.74–3.97 turtles/ha) (*Kapfer, Munoz & Tomasek, 2012*) similar to our estimates of three-toed box turtle densities in Missouri. Other studies reported much higher estimates of box turtle density. *Wilson & Ernst (2005)* reported density estimates of 16 turtles/ha in a population of eastern box turtles in Virginia at a study site with a mix of deciduous woodland, brush, and open field habitat comparable to Prairie Fork. Estimates were achieved using mark–recapture methodology and may reflect a difference in density estimates between distance sampling and mark–recapture techniques.

Habitat management and rates of disturbance differ between our two study sites and may influence overall survival in both populations. The frequent use of mowers and prescribed fire for prairie restoration is a potential source of mortality at Prairie Fork. Mortality of wood turtles (*Glyptemys inscultpa*) and eastern box turtles due to crushing by mower tires can be as great as 46% in agricultural field habitat (*Erb & Jones, 2011*). We found two turtles with extensive carapace damage at Prairie Fork; however, the amount of carapace damage in the two populations did not differ. Overall carapace mutilation at both of our study sites was moderate to low (Fig. 3). Increasing the height of mowing decks seems to be a beneficial management strategy for minimizing injuries to turtles at this site. Fire and agricultural machinery caused a 64% decrease in a *Testudo hermanni* population (*Hailey, 2000*). Similarly agricultural machinery killed 20% of a wood turtle population in southern Quebec and was the source of all of the anthropogenic mortality in a population of eastern box turtles in Delaware (*Saumure, Herman & Titman, 2007*; *Nazdrowicz,*

*Bowman & Roth, 2008*). *Stickel (1978)* noted steady declines in numbers of box turtles in a study spanning 30 years. Although the site surveyed in this population was largely managed as natural woodland, development and construction in areas surrounding the site were proposed as a cause of this steady decline in population size (*Stickel, 1978*). *Schwartz & Schwartz (1974)* noted a similar decline in their studied population of three-toed box turtles over an eight-year study period, and proposed small but continuous changes in habitat quality as a driving force behind this decline. Although we did not estimate survival or long-term population trends, turtle density and limited carapace damage suggests management activities are accommodating box turtle use of Prairie Fork.

Our observation of fewer adults in age class 3 within Prairie Fork may be explained by decreased survival of adults crossing a road that separates Prairie Fork and a nearby forest habitat southwest of the site, or by turtles in age class 3 remaining in the forest patch southwest of Prairie Fork. High rates of road crossing mortality may lead to avoidance of road structures over time causing minimal movement of individuals between two habitats (*Shepard et al., 2008*). Turtles at Baskett used forest edges and openings in contrast to turtles at Prairie Fork that used linear patches of available forest (*Rittenhouse et al., 2007*; *Rittenhouse et al., 2008*). These patterns appear similar to those described by *Currylow, Macgowan & Williams (2012)*, where box turtles were found to make short but frequent movements between forested habitat and open areas created by timber management. The population of three-toed box turtles sampled at Prairie Fork may experience higher rates of immigration as turtles migrate from the nearby forest habitat southwest of the site into open areas at Prairie Fork for nesting (Fig. 1). Females typically move greater distances daily than males and females travel into nesting sites (*Iglay, Bowman & Nazdrowicz, 2007*), and thus we expected more female box turtles in Prairie Fork than Basket. However, the sex ratio at Prairie Fork was skewed towards males at Prairie Fork (2.38:1) and equal at Baskett (1:1). The presence of males at Prairie Fork indicates that both nesting females and males are crossing the road and using the open, old field habitats at Prairie Fork.

Discernible annuli on turtle carapaces may be impacted by differences in habitat quality and/or the availability of food resources alter growth rates (*Gibbons, 1967*; *Brown, Bishop & Brooks, 1994*; *Wilson, Tracy & Tracy, 2003*; *Dodd & Dreslik, 2008*). A lack of adequate food resources or otherwise poor habitat quality may cause growth rings to grow closer together, which may result in the researcher underestimating turtle age based on annuli counts alone (*Aresco & Guyer, 1998*). Carapace length was not affected by the interaction between minimum age and site, which indicates that turtle growth rates are similar between the two sites.

Using a double observer method improved our density estimates for three-toed box turtles at Prairie Fork where vegetation was dense but provided negligible improvement to our estimate of density at Baskett. We expected confidence intervals around density estimates to decrease with a second observer, but this decrease only occurred at Prairie Fork. Probability of detection increased at Prairie Fork when increasing from 1–2 observers and the associated error for probability of detection at both Baskett and Prairie Fork decreased with 2 observers (Table 1). We achieved larger estimates of density when running a double observer analysis of distance data for both populations of turtles (Table 1)

suggesting the potential for under-estimation of turtle abundance when using single observer methodology. Our probabilities of detection were still markedly greater than those reported by *Refsnider et al. (2011)* ($P = 0.03$) for ornate box turtles (*Terrapene ornata*) in northwestern Illinois, even when only one observer was responsible for turtle sightings. While additional observers may increase overall numbers of observations and detection probabilities, *Buckland et al. (2001)* caution that in utilizing many observers the researcher runs the risk of double counting (driving animals from one transect to the another within one sampling period), or otherwise creating enough disturbance to impact the movement of individuals and the distance at which they are detected. *Refsnider et al. (2011)* suggest that detection probabilities may improve when visual encounter techniques are combined with other methods of detection. *Couturier et al. (2013)* suggest that capture–mark–recapture techniques may be preferred over distance sampling where detection probabilities are low and highly variable based on activity levels. *Fewster & Pople (2008)* suggested a combination of the 2 methods may increase accuracy while eliminating shortcomings of both methods. We did not achieve enough recaptures over the course of the study to perform a mark–recapture analysis.

Our estimates of density and observed age structure fall short of the survival and reproduction information needed to draw conclusions about trends in population size and long-term persistence. They do, however, provide insight into the current status of two populations persisting at the interface between forested and prairie landscapes. Long-term monitoring of both turtle populations is needed to determine whether the structures of these populations are shifting over time in relation to habitat management and whether human intervention is required to ensure the persistence of three-toed box turtle populations in this area.

## ACKNOWLEDGEMENTS

We thank T Bonnot, W Hoffman, D Lesmeister, D Lillard, J Strong, and N Van Dyke for field assistance, as well as several volunteers who searched for turtles. H. Pat Jones and M. Van Dyke were gracious hosts and facilitated access to additional properties. J Vokoun and M Evans provided insightful comments that improved the manuscript.

### Funding

Funding and logistical support were provided by Prairie Fork Trust Fund, Missouri Department of Conservation, and University of Missouri. The funders had no role in study design, data collection and analysis, decision to publish, or preparation of the manuscript.

### Grant Disclosures

The following grant information was disclosed by the authors:
Prairie Fork Trust Fund.
Missouri Department of Conservation.
University of Missouri.

## Competing Interests

The authors declare there are no competing interests.

## Author Contributions

- Kelly M. O'Connor analyzed the data, wrote the paper, prepared figures and/or tables, reviewed drafts of the paper.
- Chadwick D. Rittenhouse conceived and designed the experiments, performed the experiments, contributed reagents/materials/analysis tools, reviewed drafts of the paper.
- Joshua J. Millspaugh conceived and designed the experiments, contributed reagents/materials/analysis tools, reviewed drafts of the paper.
- Tracy A.G. Rittenhouse contributed reagents/materials/analysis tools, wrote the paper, reviewed drafts of the paper.

## Animal Ethics

The following information was supplied relating to ethical approvals (i.e., approving body and any reference numbers):

University of Missouri Animal Care and Use Committee approved this research (Protocols 3629 and 4291).

## Supplemental Information

Supplemental information for this article can be found online at http://dx.doi.org/10.7717/peerj.1256#supplemental-information.

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
