# Peer review of "Demographics and density estimates of two three-toed box turtle (Terrapene carolina triunguis) populations within forest and restored prairie sites in central Missouri"

_PeerJ, doi:10.7717/peerj.1256_

## Round 0.1 · original submission · Major Revisions

Your paper could be an important contribution to the knowledge of the Three - Toed Box Turtle but needs to be reviewed in depth prior to publication. Methods and figures need to be clarified (see suggestions of the reviewers). The results need more work, remember that any colleague should able to repeat the tests and obtain the same results. Both Reviewers have made valuable comments, follow them carefully.

·

Basic reporting

1) The first sentence of the intro has little to do, apparently, with the rest of the paper. Yet the second sentence gets exceedingly specific. The paper is generally clear and smooth, but this first paragraph could use some help.

2) Figure 2 is a little hard to interpret because the scales vary. Making both the X and Y axes identical for all four graphs would help. Adding a fifth panel which overlays all the curves (using different types of lines and a key) would make direct comparison easy.

3) Figure 2- Why does the probability of detection sometimes exceed 1? This result is counter-intuitive and difficult to interpret and needs to be explained somewhere.

4) Figure 3- The legend explains something about age classes which is not relevant to the figure.

5) Figure 4- These histograms are interesting. At first I thought that PF had a smoother curve indicating better sampling. Note that BA goes from ten 10yo's to three 11yo's to sixteen 12 yo's to two 13yo's while PF has smoother changes between years. However then it occurred to me that these jagged age distributions could be “real” in the sense that they may reflect year-to-year variation in survival (for example a fire in one year could wipe out most of the yearlings and a gap would move its way up the age distribution every year as that cohort gets older. So my question is, how do you interpret the differences in the smoothness of these two curves? Is one site better sampled than the other? Do they reflect detectability? Or differences between the sites' histories?

6) Figure 4- the two populations look very similar in their ranges, it may help to show the averages on this figure to emphasize the differences.

7) Figure 5 legend- It should be noted here whether the sites are different statistically.

8) A graph of carapace length vs. age would be useful and interesting. A corresponding analysis of variance of carapace length (response variable) with age (covariate) and population (factor) and the age by population effect (interaction) would permit the analysis of growth rates. Differences in growth rates between populations would be really cool!

9) Line 264 Herpetological is misspelled. then it occurred to me that these jagged age distributions could be “real” in the sense that they may reflect year-to-year variation in survival (for example a fire in one year could wipe out most of the yearlings and a gap would move its way up the age distribution every year as that cohort gets older. So my question is, how do you interpret the differences in the smoothness of these two curves? Is one site better sampled than the other? Do they reflect detectability? Or differences between the sites' histories?

Experimental design

10) Figure 1 suggests the habitats differ not just in amount of open fields, but in the amount of coniferous forest. Is it possible to quantify and report these vegetative differences? This seems especially important to understanding the differences in detectability of turtles (as well as their ecology).

Validity of the findings

Major points-

11) The biggest question I have about the entire study is how can the estimates of population density be so different for the two populations when the number of captures is so similar? I assume your effort was similar between the sites, but it's not clear to me if this is true. This difference in density estimates seems to be completely reliant on the probability of detection. I don't understand how these probabilities are calculated and why they are so different for the two populations. Since this appears to be the driving force behind the main points of the paper, this needs careful attention and explanation. Relevant to this discussion is the effort put into finding these turtles. Could the amount of time and/or distance covered/number of transects be included in Table 1? How does the understory thickness and habitat affect the detectability of turtles? How do the different models determine probability of detection? Does it vary greatly between them in results or in underlying assumptions?

12) The transects are not established randomly. The ones in prairie fork seem especially carefully placed to occur within the deciduously forested parts of the area. When calculating the density of the turtles in the area are you assuming that the transects sampled are representative of the whole area? If so, there is a major problem. Why did you not sample the open field areas? If the answer is that you knew there weren't as many turtles there, you seem to have an additional problem.

Minor points-

13) Line 67- How was estimated encounter rate calculated?

14) L78- Was the transect sampling time of day and weather consistent? Can it be specified or included in the analysis?

15) L86- Carapace cervical length is confusing. Cervical means neck and carapace means the dorsal shell. Given the values, I infer CCV to reflect mid-sagittal carapace measurements. If so, please rename your measurement or define it clearly.

Additional comments

Generally I found this work interesting and think the majority of comments above are relatively minor (and some are just suggestions). The major points (especially #11 and 12) may simply require more in depth explanations, but if the sampling methods are not representative of the sites, then a reinterpretation of the data is necessary and conclusions could change. In any case, I hope you found these comments useful and best wishes for a fruitful revision!

Reviewer 2 ·

Basic reporting

The language is poor enough that the authors should have a third party review it for style before resubmission. The manuscript also lacks sufficient background and context. It is not clear to me that all of the data, i.e., the capture data, is published somewhere that I could rerun their models.

Experimental design

The statistics need some work (see comments to author, below)

Validity of the findings

If statistics edits are made and results are similar, conclusions could be valid.

Additional comments

General comments
The manuscript could be an important contribution to the literature. However, it requires some major revisions. First, the writing is careless enough that it is distracting. For example, the sentence on line 12-13 has two tenses in it where this is not appropriate. I recommend that the manuscript be proof-edited by a third party before resubmission. Second, the manuscript ignores lots of data already in the literature. For example, there is a wealth of data on expected population densities in Terrapene spp. box turtles, but the authors largely do not compare their findings to those of others (e.g., see review in Dodd 2001). The other findings are likewise poorly placed into the context of the literature, lessening the importance of the present work. I recommend major revisions.
Specific comments
Title: Revise – the current title reflects only the findings about density. Perhaps something about population demographics and effects of survey type.
Abstract – The abstract should mention your findings about survey type and injuries, as these were main goals of the manuscript.
Introduction – Though density and age structure are important foci of the manuscript, they are not introduced here. Why are you studying these parameters? What are their conservation implications? What about survey type? Why test different techniques? Basically, anything you mention as a study objective should be set in context for the reader.
Line 13 – Both “carolina”s should be lower case.
Line 13 – Change “increase” to “increased”. Double-check tense throughout.
Line 21 – This line needs a citation to substantiate it.
Materials and Methods
Lines 64-66 – Specifically, how was Buckland et al. use to determine the predicted encounter rate and transect length?
Lines 72-73 – Justify the use of multiple samples over time to get sufficient coverage. This appears to violate the assumption that while running a transcript, each individual observation is independent. Repeated sampling might run the risk of sampling the same individuals more than once.
Lines 80-81 – I agree that the number of rings is at best an age minimum, but you need to better justify from the literature the use of these rings as annuli. There is also a strong argument that, like tree rings, growth rings might be small or absent in years of poor resource acquisition or injury or other factors. If that’s more likely in a suboptimal habitat (such as you seem to suggest is present at one or both of your sites), then a signature of significant age structure difference might alternatively be a signature of different habitat quality.
Lines 91-93 – After this sentence, comment on how you controlled quality across observers. Was there some attempt to agree on the scoring? Were they repeatable across time or did they improve with practice?
Line 96 – “program DISTANCE” requires a citation.
Lines 99-100 – The use of “visual histogram analysis” for outlier is probably not valid given there are more formal tests for outliers. Also, you need to justify whatever you planned to do with outliers. You cannot just omit them without cause (being an outlier is not by itself a reason to disregard a datum).
Lines 110-111 – See above comment.
Results
Line 121 – Were these all unique captures? How many recaptures? If you had recapture data, were you able to calculate a census population size?
Line 126-128 – Was this one of the top three models?
Line 133-134 – As above: how good was this model? It looks like the model choice is somewhat arbitrary.
Discussion
You estimate a priori what the encounter rate was (line 65). Did your results match that? Why or why not?
Line 156 – Subject-verb agreement: change “is” to “are”.
Lines 159-161 – You statement is inappropriate as you did not test this. You measured demographic data, and you can suggest that observed differences may be due to difference in management practices, but this statement is too bold.
Lines 164-166 – You should note that the null is also observe: there is sometimes no difference in habitat used for nesting (e.g., see Dodd et al. 1994).
Line 171 – This paragraph (or a new one) should also discuss your findings in light of the body of work or Schwartz and Schwartz, which was also in Missouri three-toed box turtles. Perhaps Stickel’s work too. And Williams and Parker 1987.
Line 208-209 – Again, the literature does not always support this. For example, see Currylow et al. 2012 and Dodd et al. 1994.
Line 222 – If two observers was better than one, you must also admit that three or more might be even better and necessary to adequately census box turtle populations.
Line 227 – You should also discuss how your detection rates compare to that of Refsnider et al. 2011.
Figure 1 – The property boundaries and transcripts are important and so should be represented in the legend. They should also be distinct.
Figure 1 – The two maps should have identical grays for each landcover type, should have identical land covers represented, and identical orders in the legends. Also, “Open” is too inclusive and different grays (or patterns) should be used for roads and other open habitats, since they are important to the thesis of the manuscript.
Figure 4 – The x-axis needs units.
Table 2 – This is a clumsy table. Means and SDs can be in one column. n for PF should be in a column before Mean PF.

---

## Round 0.2 · Minor Revisions

I am sorry that it has taken longer than expected to respond to your revision. The original editor was unable to make the decision due to illness and I was asked to replace her.

The reviewer has indicated that your manuscript has been substantially improved but has provided a number of additional helpful suggestions.

I was surprised by the reviewer's comment on the term 'methodology'. According to on-line dictionaries, the first definition is 'a body of methods, rules, and postulates employed by a discipline: a particular procedure or set of procedures' and the second definition is the study of methods. I think that your use of the term fits the first definition and can remain as written, if you prefer.

I do have a concern about one issue that was apparently not raised by the previous reviewers or editor. The potential problem involves your repeated sampling of the same transects. I am not particularly sophisticated in statistics, but I think it is clear that these repetitions would not be independent and that treating them as such, rather than using some repeated-measures procedure, would inflate the degrees of freedom. You must be much clearer about how many times you repeated the same transects and how you treated these repetitions in your statistical analysis. Your statement about multiplying the length of the transect by the number of times visited does not make sense to me as a statistical approach. If, in fact, you treated the repetitions as if they were independent replicates, your analysis is likely to be incorrect. You should check with a statistician to determine whether your approach is valid and, if not, to obtain advice on how to correct the analysis.

Methods - The implication is that rain caused you to postpone but never to skip a sample (L113). Is that correct? If so, be a bit more explicit and provide the number of samples. Later comments seem to imply that there may have been different numbers of repetitions of different transects.

L114. Specify the temperature that you considered could restrict activity (L114).

Results L186. The reference to competing models is unclear, because all your models were competing in all cases. If you mean that it was one of a set of well-supported models, state this more directly. Note that the evidence ratio provides a quantitative way to compare the similarity or difference of any two models in AIC.

Discussion: Is it worth making the explicit point that you found similar numbers of turtles in similar numbers of transects at two sites but that the more than two-fold difference in densities was only realized by using a method that accounts for visibility? I have experience in transects on coral reefs but have not used this method, and the effect is quite striking to me.

Grammar and clarity

Abstract L17. 'Replication . . . is needed . . .' (agreement between subject and verb)
L63. Colon should not used after verb 'to be'
L77-79. This sentence is unclear. If I understand what you mean, perhaps a set of commas is needed: '. . . and, outside of the area sampled for turtles, is . . .'
L84. Separate independent clauses with a comma '. . . streams, and a large . . .'
L85. 290 ha is redundant with L82
L89. research-oriented (hyphen needed for compound adjective)
L96-98. Wordy sentence. Consider: 'See Buckland et al. (2001) for a detailed explanation.'
L105, 254, 284. Therefore, however, (adverbial conjuncts such as therefore and however are followed by a comma; check for this throughout your manuscript)
L106. Awkward sentence. Consider: 'Therefore, our density estimates apply only within the forested areas of our study sites.'
L111. The meaning of 'desired length of sampling' is rather unclear; do you mean 'desired sample area' or 'desired total transect length'? Note that you have told readers the desirable number of transects but not their length, so this statement is not completely logical.
L129, 132. Redundant. No need to repeat caliper reference.
L201-204. Text and table are redundant. Present the details in only one place.
L209. For greater clarity, specify the response variable instead of using the general term.
L215-216. Redundancy. You do not need to specify the densities in this general statement as well as later in the Discussion. The information is more relevant below where you elaborate on their relation to the published literature.
L230. 'with regard to' (incorrect use of regards). 'limited in habitat suitable for nesting' would be just as clear.
L241-243. 'contained densities of . . . turtles . . . similar to our estimates of three-toes . . .' (move similar to be adjacent to the preposition that modifies it).

Figures. It is not usual to include gaps/multiple paragraphs in figure captions. Revise to form a single paragraph.
Fig. 1. I had trouble distinguishing 5 shading scales. Consider using a different pattern (e.g., stippling) for water and food plots. Also, in the legend, the white rectangle for boundary seems to imply that the boundary is designated by white. If you mean that the type of lines around the box match the boundary, you should use a line, as you did for transect locations. A wider line for the boundary might be clearer.
Fig. 2. I don't think the panel designations need to be in bold. At any rate, all four should be the same.
Fig. 4. The parentheses showing abbreviated names for the study sites do not correspond to the initials used for this purpose in the figure. The second paragraph with information about the means and sampling issues belongs in the Results not the caption. Be sure to indicate whether the variation refers to SD, SE, or CL here and elsewhere in the manuscript.
Fig. 5. Modify as suggested for Fig. 4. Also, remove 'captured'. The word does not provide additional information and is grammatically incorrect: the turtles were not captured in age classes.
Table 1. The heading is incomplete.
Table 2. Avoid two-paragraph table heading. Consider providing a separate column for statistical significance.

·

Basic reporting

No comments

Experimental design

No comments

Validity of the findings

No comments

Additional comments

Review 2 of O’Connor et al. for PeerJ

Generally, the MS is improved and much clearer. A few issues exist that should be fairly straight forward to address.

Abstract- The conclusion that, “Replication of this study across more sites, as well as across time, are needed to fully understand the implications of habitat management on box turtle populations.” is weak. Is there no larger picture take away from this study? The big picture to me seems to be that the efficacy of 1 vs. 2 observers depends on the habitat.

L29: The fitness of a population is not clear. Are you talking about group selection? Population growth rate? Fecundity?

L63: The objectives seem to downplay the way the intro leads us to the important part of the study which is comparing one vs. two observers.

L85: Significant digits for estimated cover should be consistent and the total should ideally add to 100%

L96-8: It seems that an example would be useful here. Eg. Detection probability decreases more quickly with distance in dense cover than in open areas (see Buckland et al. 2001). (ps. Methodology is the study of methods.)

L111: The text implies that every transect was sampled every week for approximately 10 weeks. The number of samples per transect should be made explicit. Also, does the replacement of independent spatial replicates with repeated samples of some transects assume that all times are equal? Should we expect that mid-May and late July have different detectability due to vegetation growth as well as turtle behavior and movement patterns due to breeding, resource levels and environmental temperature?

L137: replace insure with ensure

L192: If I understand the data and analyses, the distance of turtles from the transect should be lower in the Prairie than Baskett. Reporting the average (and s.d.) distance from transect in Table 1 might substantiate this and illustrate the effects you are describing here.

L192: the reference apparently should be to Table 1 not table 2.

L197: It is not clear why these two turtles are being singled out.

L201: the ratio could be important for functional population estimates (as stated in the introduction). However, this is not discussed (L283). Dividing the population density estimate of PF by the sex ratio (4.14/2.38) produces a value (= 1.74) virtually identical to the population estimate of BA (1.85). Does this mean that the two sites have similar densities of females and therefore similar effective population densities (ie. number of reproductive individuals/ha).

L201-4 these data should either be in the graph or described in the text, but it is redundant to put the same data in both places.

L233: This sentence is redundant with a previous one (L215). Also, your estimates do not "overlap somewhat" with the one you reference. It is outside their range. It could be said to be similar.

L256: Raising the mower deck would not protect the turtles from tire damage. I would expect tire damage to appear as cracked shells and blade damage to be evident by "scalping". Do you have evidence that the damage is occurring by one or the other of these methods.

L284: I do not understand the logic or meaning of this concluding sentence.

L286-7: This sentence is incomplete and lacks agreement.

L292-4: The ANCOVA you performed on carapace length with minimum age and site is a test of this. The conclusion should be that while carapace size increases with minimum age (yay) turtles at the two sites grow at the same rate.

L308: delete the extra "the"

L315: replace "methodologies" with "methods"

Figure 2: I would prefer to see this as a two panel figure with a panel for each site and each panel showing a line for both single and double observers. This would make the effects much easier to see and compare.

Table 1: there is a hanging open parenthesis in the title.

Table 2: in contrast to what the title says "All measurements excluding carapace mutilation (CMI) were significantly different between turtles at BA and PF (all p-values <0.05).", the p-value for Age Class is equal to 0.05. This is also a strange bit of text to be in the title of a table and sounds more like Results.

Similarly, for Figure 5 legend: Since the alpha is 0.05 and the p value is = 0.05 then the p-value is not less than the alpha and it is not statistically significant, correct?

---

## Round 0.3 · accepted · Accept

I consider this manuscript now suitable for publication.